# Networks are Slacking Off: Understanding Generalization Problem in Image Deraining

**Jinjin Gu**[1,2]   **Xianzheng Ma**[1]   **Xiangtao Kong**[1,3]   **Yu Qiao**[1,3]   **Chao Dong**[1,3,*]

[1] Shanghai AI Laboratory   [2] The University of Sydney
[3] Shenzhen Institutes of Advanced Technology, Chinese Academy of Sciences
jinjin.gu@sydney.edu.au, xianzhengma@pjlab.org.cn
{xt.kong, yu.qiao, chao.dong}@siat.ac.cn

## Abstract

Deep deraining networks consistently encounter substantial generalization issues when deployed in real-world applications, although they are successful in laboratory benchmarks. A prevailing perspective in deep learning encourages using highly complex data for training, with the expectation that richer image background content will facilitate overcoming the generalization problem. However, through comprehensive and systematic experimentation, we discover that this strategy does not enhance the generalization capability of these networks. On the contrary, it exacerbates the tendency of networks to overfit specific degradations. Our experiments reveal that better generalization in a deraining network can be achieved by simplifying the complexity of the training background images. This is because that the networks are "slacking off" during training, that is, learning the least complex elements in the image background and degradation to minimize training loss. When the background images are less complex than the rain streaks, the network will prioritize the background reconstruction, thereby suppressing overfitting the rain patterns and leading to improved generalization performance. Our research offers a valuable perspective and methodology for better understanding the generalization problem in low-level vision tasks and displays promising potential for practical application.

## 1   Introduction

The burgeoning progress in deep learning has produced a series of promising low-level vision networks that significantly surpass traditional methods in benchmark tests [10, 30, 12]. However, the intrinsic overfitting issue has prevented these deep models from real-world applications, especially when the real-world degradation differs a lot from the training data [20, 31, 33, 34]. We call this dilemma the generalization problem. As a traditional and important low-level vision task, image deraining also faces a severe generalization problem. Existing deraining models tend to do nothing for the rain streaks that are beyond their training distribution, see Figure 1 for an example. The model trained with synthetic rainy images cannot remove rain streak for images with different rain patterns as the training data.

Despite its significance, the generalization problem in the context of deraining tasks is not comprehensively explored in the existing literature. Understanding the underlying causes of these generalization performance issues is imperative to propose effective solutions. However, analyzing generalization in a low-level task is far from straightforward, as it is not a simple extension of the generalization research in high-level vision tasks. This paper aims to take the pioneering step towards a more profound understanding of this challenge.

---

[*]Corresponding author.

37th Conference on Neural Information Processing Systems (NeurIPS 2023).

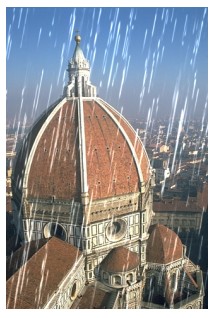 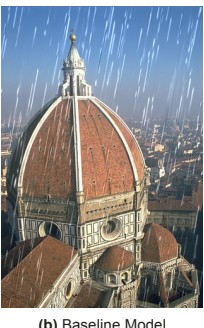 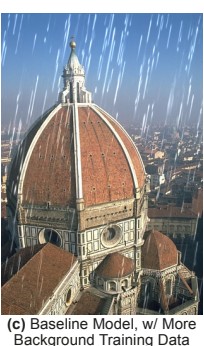 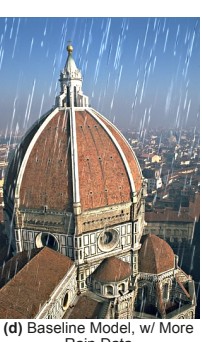 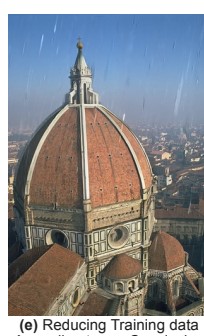

**(a)** Input Rainy Image     **(b)** Baseline Model     **(c)** Baseline Model, w/ More Background Training Data     **(d)** Baseline Model, w/ More Rain Data     **(e)** Reducing Training data According to our Conclusion

Figure 1: The existing deraining models suffer from severe generalization problems. After training with synthetic rainy images, when feeding **(a)** an image with different rain streaks, its output **(b)** shows a limited deraining effect. Two intuitive ways to improve generalization performance, including **(c)** adding background images and **(d)** adding rain patterns, cannot effectively relieve the generalization issue. In this paper, we provide a new counter-intuitive insight: **(e)** we improve the generalization ability of the deraining networks by using *much less* training background images for training.

As a typical decomposition problem, image deraining utilizes a relatively simple linear superimposition degradation model. When the network fails to generalize, the rain streaks persist, and the image background remains unaffected. This intuitive and quantifiable phenomenon is well-suited for our study. Contrary to previous works that rely only on overall image quality evaluation, we propose to decouple the deraining task into rain removal and background reconstruction. These two components are then separately analyzed. A key motivation behind this approach that the unsatisfactory performance can be attributed to "the unsuccessful removal of rain streaks" or "the poor reconstruction of image background". Without distinction, an image with successful rain streak removal but poor background reconstruction will also have bad quantitative results. Given that the generalization problem in the deraining task is mainly related to the removal of rain streaks, we argue that those results that can remove rain will be more enlightening, regardless of the background reconstruction effect. Our research method allows us to minimize the impact of other less relevant factors on the effectiveness of rain removal.

In this paper, we show that the generalization problem arises when the network overfits the degradation, *i.e.*, the rain patterns present in the training set. One significant factor contributing to this outcome is the inappropriate training objective. We start our research with the most fundamental element in formulating the training objective – the training data. Numerous studies have attempted to enhance real-world performance by increasing the complexity of training data. This approach originates from a natural but unproven "acknowledgement" that augmenting the quantity of training data can rectify the generalization problem. This "acknowledgement" has also permeated the deraining community, suggesting that a network trained with a more diverse training set (both in terms of background images and rain streaks) can better generalize to unseen scenarios. However, this approach does not effectively address the generalization problem in deraining. We argue that this issue arises precisely because the network is provided with an excess of background data during training. Consequently, the model fails to learn to reconstruct the image content but instead overfits the degradation. We arrive at some counter-intuitive conclusions by employing our analysis method, which separately measures background reconstruction and deraining effect.

**Our key findings.** We find that deep networks are slacking off during training, aiming to reduce the loss in the quickest way. This behaviour leads to poor generalization performance. The improper objective set for training is one of the primary contributors to this issue. Our finding indicates that deep networks exhibit a tendency to learn the less complex element between image content and additive degradation.

Specifically, the network naturally learns to overfit the rain streaks when the background complexity is higher than the rain complexity. However, when rain streaks in real-world scenarios deviate from the training set, the network tends to disregard them. At this point, the network will not remove rain streaks from the image and exhibit poor generalization performance. Conversely, training the model using a less complex background image set demonstrates superior generalization ability, as illustrated in Figure 1 (e). This is because that when the complexity of the training image background is less than that of the rain patterns, the network will again take a shortcut to reduce the loss. In this case, the network learns to reconstruct the background rather than overfitting the rain streaks. When faced

with rain streaks outside the training set, the network takes priority to reconstruct the background, thus avoiding failure caused by the inability to recognize new rain streaks. It is also important to note that the model's performance is determined by the removal of rain and the quality of background reconstruction. Reducing the background complexity of the training data might inevitably lead to subpar reconstruction performance. Nonetheless, our results revealed that a model trained with just 256 images can already handle most image components effectively. These counter-intuitive phenomena have not been revealed in previous literature.

**Implication.**   Our results highlight the critical role of the training objective in determining a model's generalization ability. An inappropriate or incomplete training objective creates a loophole for deep networks to "slack off". While we anticipate that the network will learn the rich semantics inherent in natural images, it is often overlooked that the model can also achieve learning objectives through shortcuts. This results in poor generalization performance of low-level vision models. Our findings also highlight that a model with good generalization ability should learn the distribution of natural images rather than overfitting the degradation.

## 1.1   Related Works

This research primarily pertains to the field of deraining. However, unlike most of the existing deraining research, we do not propose new network structures, loss functions, or datasets. Our objective is to analyze and understand the generalization problem within the context of the deraining task. Due to space constraints, reviews of deraining works are provided in Appendix A.1. We then review previous works focusing on interpretability and generalization in low-level vision.

Deep learning interpretability research aims to understand the mechanism of deep learning methods and obtain clues about the success/failure of these methods. We are not convinced to move forward in the right direction without a deep understanding of these working mechanisms. The research on deep learning interpretability follows a long line of works, most focusing on the classification task [47, 48, 46, 49, 76, 36]. Low-level vision tasks have also embraced great success with powerful deep learning techniques. There are also works on interpretability for these deep low-level networks [19, 61, 37, 44]. For the generalization problem in low-level vision, these problems often arise when the testing degradation does not match the training degradation, *e.g.*, different downsampling kernels [20, 31, 27, 74] and noise distributions [22, 7]. The existing works either develop blind methods to include more degradation possibilities in the training process or make the training data closer to real-world applications. Only a little work has been proposed to study the reasons for this lack of generalization performance [33, 34]. More details of these previous works can also be found in supplementary material. No research has attempted to investigate the interpretation of the training process of low-level vision networks, especially from the perspective of the generalization problem.

## 2   Analysis Method

We aim to investigate how different training objectives influence network behaviour and generalization performance. Before detailing our observations, we will outline our experimental designs and quantitative analytical methods.

### 2.1   Construction of Training Objective

The training data and the loss function jointly determine the training objective of a deep network. We set various training objectives to observe the changes in the generalization performance of different deraining models. As shown in Figure 3 (left), a rainy image $O$ can be modelled using a linear model $O = B + R$, where $B$ is the image background, and $R$ is the additive rain image. We change the training objectives with different background images and rain streaks.

**Background Images.**   Typically, image backgrounds are sampled from street view images [17] or natural image datasets [43, 1], as these images are close to the application scenarios of deraining. In literature, previous works [13, 71] claim that the model can learn the prior knowledge of reconstructing these scenes by training on a large number of such background images. We break this common sense by constructing different training background image sets from the following two aspects.

First, we change the number of training background images. The amount of training data partially determines the complexity of network learning. We argue that as too many background images are provided for training, the model cannot faithfully learn to reconstruct the image content but turns to overfit the degradation patterns. We reduce the complexity of the background images to see how the network behaviour changes in extreme scenarios. In our experiments, we use 8, 16, 32, 64, 128,

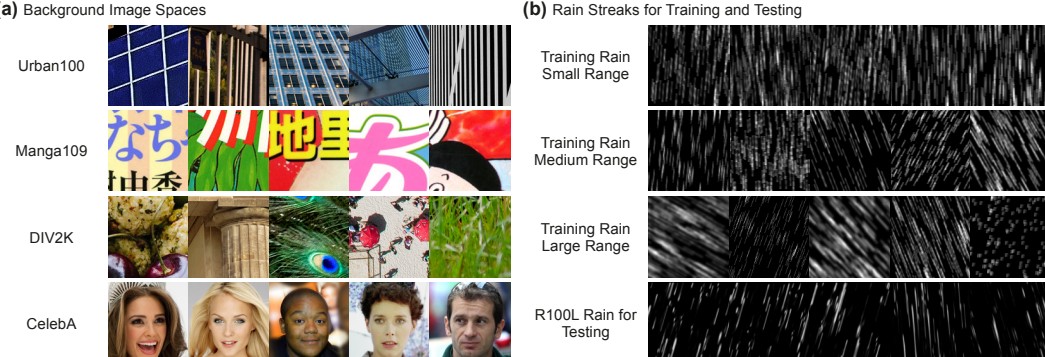

Figure 2: **(a)** Background images from different image categories. It can be seen that the structure of the face image (CelebA) is relatively complex. Natural images (DIV2K) contain natural textures and patterns. The patterns in Manga109 and Urban100 are artificially created – Manga images have sharp edges, while Urban images contain a lot of repeating patterns and self-similarities. **(b)** Different synthetic rain streak distributions that were used in our experiments.

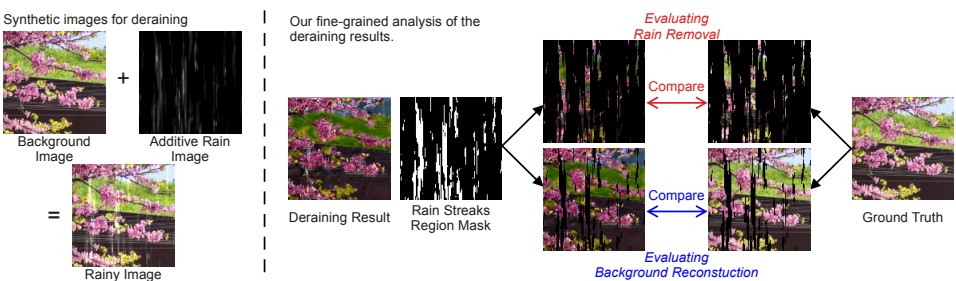

Figure 3: **(Left)** The illustration of the rainy image synthesis. **(Right)** Our fine-grained analysis of the deraining results.

256, 512, and 1024 background images[2] to build the training datasets, respectively. We also use a large number of images (up to 30,000) to simulate the typical situation when the image background is sufficiently sampled.

In addition to the data scale, the image content will also affect network learning. For example, it is easier for the network to fit the distribution of images with regular patterns. A face image containing both short- and long-term dependent structures is more complex than a skyscraper with only repeated lines and grids [2]. We select image distribution as the second aspect of our dataset construction. We sample from four image datasets that are distinct from each other: CelebA (face images) [35], DIV2K (natural images) [50], Manga109 (comic images) [38], and Urban100 (building images) [24]. Some examples of these images are shown in Figure 2 (a).

**Rain streaks synthesis.** Since it is hard to collect a large number of real-world rainy/clean image pairs, we follow the previous deraining works [16, 13] to synthesize rainy images for research. We use two kinds of rain streaks for training and testing separately. For training, we

Table 1: Different rain streaks settings.

| Range | Quantity | Width | Length | Direction |
|---|---|---|---|---|
| Small | $[200, 300]$ | $\{5\}$ | $[30, 31]$ | $[-5°, 5°]$ |
| Medium | $[200, 300]$ | $\{5,7,9\}$ | $[20, 40]$ | $[-30°, 30°]$ |
| Large | $[200, 300]$ | $\{1,3,5,7,9\}$ | $[5, 60]$ | $[-70°, 70°]$ |

use the computational model [3] to render the streaks left on the image by raindrops of varying sizes, densities, falling speeds, and directions. This model allows us to sample rain streaks from different distributions. We adopt three rain image ranges for training, where different ranges may lead to different generalization effects; see Figure 2 (b) for a convenient visualization and Table 1 for the detailed settings. For testing, we use the synthetic rain patterns in [63]. Although the rain streaks are visually similar to humans, they still pose a huge generalization challenge to existing models.

**Loss Function.** In low-level vision, the loss function is usually defined by the difference between the output image and the ground truth. We use the $l_1$-norm loss without losing generality.

---

[2]In actual training, we crop large images of different sizes into image patches of size $128 \times 128$ for training. In this article, unless otherwise stated, these quantities refer to the number of patches.

[3]The re-implementation of the PhotoShop rain streaks synthesis method. Please refer to this link.

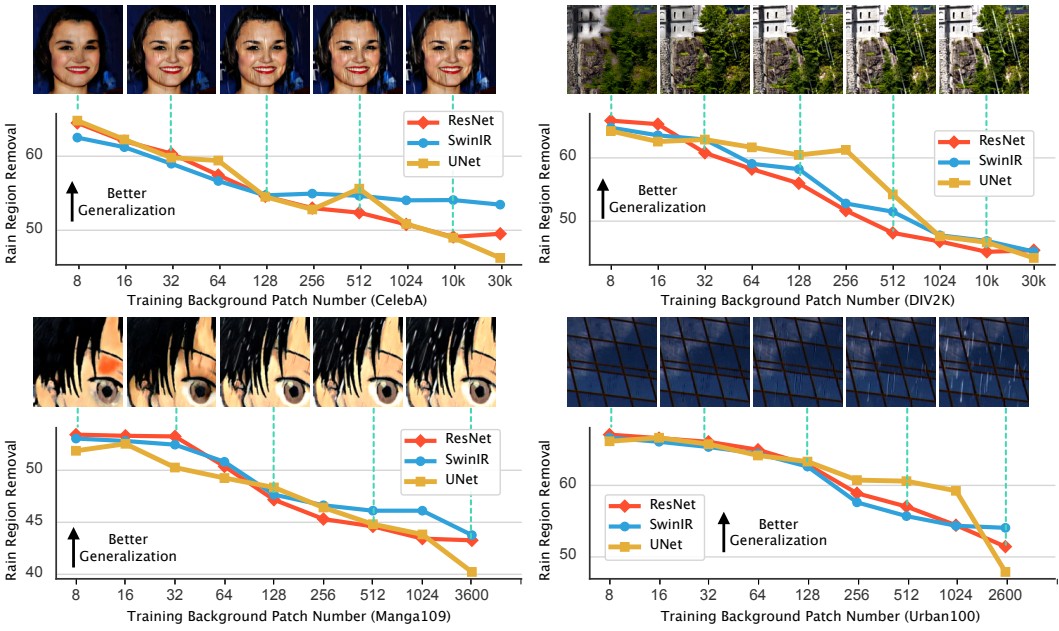

Figure 4: The relationship between the number of training background images and their rain removal performance. The $x$-axis represents the image number, and the $y$-axis represents the rain removal effect $E_R$. Higher $E_R$ means better rain removal performance. The test rain patterns are not included in the training set. Thus, the effect of rain removal at this time reflects the generalization performance. The qualitative results are obtained using ResNet.

## 2.2 Decoupling Analysis of Rain Removal Results

The standard evaluation of a deraining model involves computing similarity metrics between its output and the ground truth images [18]. However, such an evaluation strategy may lead to unfair comparison. For example, an image with perfect background reconstruction but inferior rain removal may have a higher PSNR value than that with perfect rain removal but a slightly flawed background reconstruction (*e.g.*, color shift). Drawing conclusions from such metrics may lead to biased conclusions about a model's generalization capability. Generally speaking, the generalization performance of a deraining model is mainly shown in the form of removing *unseen* rain. The background reconstruction may affect the visual effect, but it does not correlate directly with the effectiveness of rain removal. However, background reconstruction affects the quantitative result of traditional rain removal metrics. Therefore, we discuss the removal of rain streaks separately from the reconstruction of the background. We can achieve this goal with a simple mask-based decoupling method.

It can be seen from Figure 2 (b) that the pixels in the additive rain image $R$ without rain streaks should be black, while rain streaks will appear brighter. After synthesis, these black areas reflect the background area, while the brighter areas indicate the rainy regions. A perfect rain removal effect should do minimal damage to the background area and remove the additive signal from the rain streaks area. By processing $R$ to a binary mask $M$ using a threshold $t$, where $M_{[i,j]} = 0$ if $R_{[i,j]} \leq t$ and $M_{[i,j]} = 1$ if $R_{[i,j]} > t$, we can segment the output image $\tilde{O}$ into the rain streaks part $\tilde{O} \odot M$ and the background part $\tilde{O} \odot (1 - M)$. We then have two metrics:

- $E_R = \sqrt{\mathbb{E}[(\tilde{O} \odot M - O \odot M)^2]}$ gives the *Rain Removal* performance. A network with poor generalization will not remove rain streaks. This term measures the changes made by the network in the rainy regions. A higher value reflects better rain removal performance.

- $E_B = \sqrt{\mathbb{E}[(\tilde{O} \odot (1 - M) - B \odot (1 - M))^2]}$ gives the effect of *Background Reconstruction* by comparing the background regions with the ground truth. A large error in this term means poor reconstruction quality.

## 2.3 Deep Models

We summarize existing networks into three main categories in our experiments. The first category includes networks composed of convolutional layers and deep residual connections, and we use

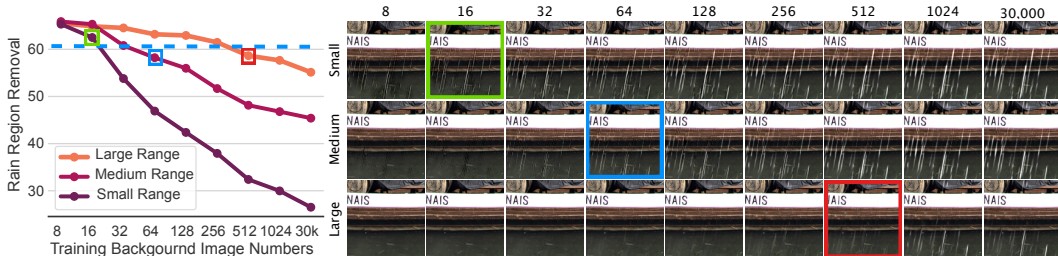

Figure 5: When trained with different rain ranges, the model exhibits different rain removal effects. The $y$-axis represents the quantitative rain removal effect. When the rain removal performance is lowered to the blue dashed line, the qualitative effect of removing rain decreases significantly. We use ResNet in this experiment.

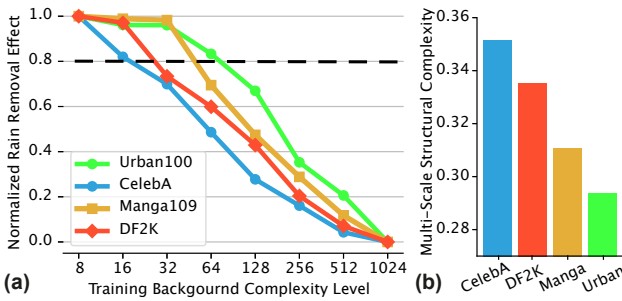

Figure 6: **(a)** The relationship between the number of training images and their normalized rain removal performance. When the $y$ value is lowered to the grey dashed line, the qualitative effect of removing rain starts to decrease significantly. **(b)** The averaged complexities of different image categories given by [2].

the ResNet [28] as a representative. The second category includes networks with encoder-decoder designs, and we use UNet [42] as a representative. UNet introduces down-sampling and up-sampling layers to extract global and multi-scale features, which have been proven successful in many deraining networks. The last category includes image processing Transformer networks. Transformer [45, 73, 75] is a new network structure characterized by self-attention operations. We select SwinIR [30] as a representative. For more training settings, please check the supplementary material.

## 3 Understanding Generalization

Next, we conduct experiments based on the above analysis method to explore the counter-intuitive phenomenon in terms of generalization in the deraining task. Our analysis includes two aspects – rain removal (Section 3.1) and background reconstruction (Section 3.2).

### 3.1 Generalization on Rain Removal

We analyze the rain removal effect on *unseen* rain streaks. Since we use different types of rain streaks for training and testing, the rain removal results can reflect generalization performance. After extensive experiments, we arrive at the following observations.

**Training with fewer background images produces better deraining effects.** In this experiment, we use the middle-level range of rain streaks and different background image sets to generate different training datasets, as described in Section 2.1. Experiments are conducted across all four categories of images. We then test the rain removal performance $E_R$ of these models. The test images utilize rain streaks proposed in [63], which are different from the rain distribution of the training set. The testing background images are sampled from the corresponding categories and differ from those in the training set. The experimental results are presented in Figure 4. Despite variations in background images and network architectures, a consistent trend is evident across all experimental results. Notably, deraining models trained on only eight images demonstrate a noteworthy ability to handle unseen rain streaks. In contrast, models trained using a large number of images exhibit an inability to address these rain streaks. *Such a phenomenon challenges traditional expectations.* Between these polarized states, the rain removal efficacy diminishes as the quantity of training images increases. Once the image number ascends to 256, the networks have predominantly lost their deraining capability. As the image number increases from 1024 to 30,000, there's minimal variance in models' rain removal performance – they all fail to remove unseen rain streaks. This trend is also reflected in the qualitative results.

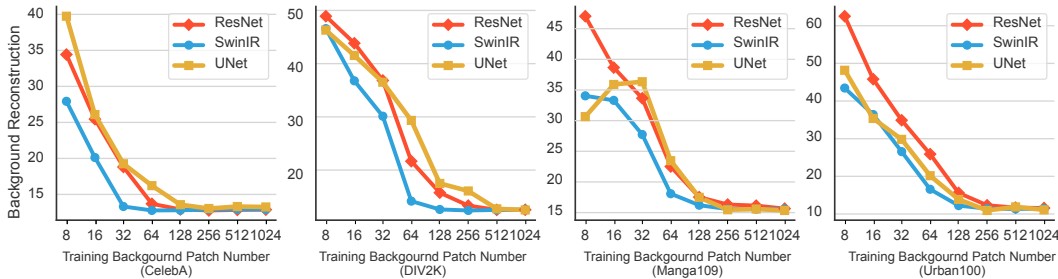

Figure 7: The relationship between the number of training images and their background reconstruction effect. For each plot, the $x$-axis represents the image number, and the $y$-axis represents the reconstruction error of the background $E_B$.

Here, we attempt to explain this interesting counter-intuitive phenomenon. Although we set the training objective as removing rain streaks from images by comparing the output to the target, the network has two alternative strategies to minimize this training loss. The first strategy involves recognizing and removing rain streaks, while the second consists of recognizing and reconstructing the image background. If the learning strategy is not specified, the network tends to gravitate towards the simpler strategy. When a large number of background images are utilized in training, learning to reconstruct backgrounds becomes significantly more complex than learning to remove rain streaks. Consequently, the network chooses to recognize and remove the rain. This decision, however, can lead to an overfitting problem: when new rain streaks differ from those used in training, the network fails to recognize and remove them. Conversely, when the background image is comprised of only a few image images, learning the background becomes easier than learning rain streaks. In this scenario, the network recognizes image components in the background without overfitting the features of rain streaks. As a result, the model demonstrates superior rain removal capability in images with unseen rain streaks.

**The relative complexity between the background and rain determines the network's behaviour.** To corroborate the abovementioned conjecture, we modify the range of the rain streaks used in training, as outlined in Section 2.1. When employing a medium-level rain range, the rain removal effect diminishes when training with 64 background images. According to our explanation, a larger rain range complicates the network's task of learning the rain pattern. As a result, the rain removal effect does not deteriorate until a larger number of background images are used for training. The experimental results are shown in Figure 5. It can be observed that the rain removal effect decreases across all three training rain ranges as the number of background images increases. When sufficient background images are used for training (30,000 images), even with a large training range for rain, the resulting model struggles to deliver satisfactory rain removal performance on unseen rain streaks. This suggests that the large rain range does not encompass our testing scenarios and cannot bring generalization ability on test images. If training with the large rain range, the network displays a significant drop in rain removal performance only when more than 512 background images are used for training. Conversely, a model trained on a small rain range cannot exhibit satisfactory rain removal even with only 16 background training image images. These results indicate that the relative relationship between the background image and rain streaks influences network behaviours. The complexity or learning difficulty of the medium-range rain is approximately less than that of 64 training background images. In comparison, the complexity of the large-range rain is approximately less than that of 512 training background images. Depending on the situation, the network tends to take shortcuts or select the easier learning pathway.

**A more complex background dataset makes learning harder for the network.** Next, we modify the category of the background images used for training and monitor the resulting behaviours of the models. We normalize the deraining effect to a range between 0 and 1 to facilitate comparison across different image categories. The results are shown in Figure 6 (a). The most intuitive conclusion is that, even when the number of training images remains consistent, different image categories can lead to different rain removal performance. For instance, in the case of CelebA images, increasing the training set from 8 to 16 images leads to a notable decrease in deraining performance. This disparity is more stark compared to natural image – increasing to 16 images does not result in a significant drop in rain removal performance. Moreover, for images sampled from the Manga109 and Urban100 categories, the rain removal performance does not exhibit a significant drop until the image number exceeds 32. According to our interpretation, the more complex image categories will prompt the

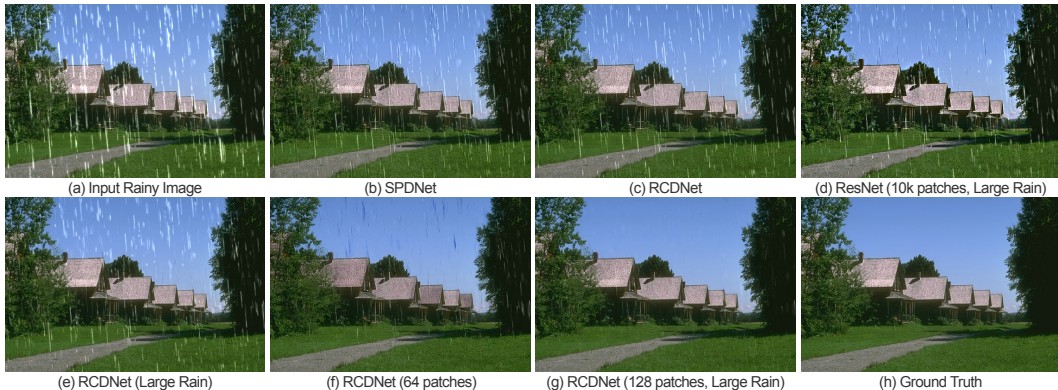

|          | (a) Input Rainy Image | (b) SPDNet | (c) RCDNet | (d) ResNet (10k patches, Large Rain) |
|          | (e) RCDNet (Large Rain) | (f) RCDNet (64 patches) | (g) RCDNet (128 patches, Large Rain) | (h) Ground Truth |

Figure 8: Visualization of the deraining results on a synthetic image. Zoom in for better comparison.

| Training Back. | Objective Range | ResNet $E_R \uparrow$ | $E_B \downarrow$ | PSNR $\uparrow$ | SPDNet [68] $E_R \uparrow$ | $E_B \downarrow$ | PSNR $\uparrow$ | RCDNet [54] $E_R \uparrow$ | $E_B \downarrow$ | PSNR $\uparrow$ |
|---|---|---|---|---|---|---|---|---|---|---|
| 30k | Medium | 31.24 | 10.79 | 25.15 | 33.63 | 5.49 | 30.51 | 26.55 | 5.41 | 28.54 |
| 64 | Medium | 53.33 | 25.02 | 20.87 | – | – | – | 45.47 | 14.78 | 25.32 |
| 512 | Large | – | – | – | 39.88 | 8.91 | 28.57 | 37.53 | 7.16 | 29.60 |
| 256 | Large | 45.64 | 16.51 | 24.30 | 38.87 | 8.03 | 29.40 | 40.40 | 8.52 | 29.08 |
| 128 | Large | 51.75 | 23.53 | 21.45 | 43.20 | 14.59 | 25.67 | 44.67 | 13.72 | 26.09 |

Table 2: Quantitative comparisons between different models. $\uparrow$ means the higher the better while $\downarrow$ means the lower the better.

model to experience an earlier performance decline as the number of training images increases. Our results suggest that the complexity of these four image categories can be ranked in ascending order as CelebA, DIV2K, Manga109, and Urban100.

The observed sequence roughly aligns with human perceptual tendencies. Face images, as represented by the CelebA dataset, exhibit strong global and local structures. While DIV2K images are abundant in local texture, they maintain a relatively simple global structure. Contrastingly, Manga images, although free of complex textures, often contain text elements and detailed edges. Lastly, Urban images are chiefly characterized by recurrent patterns, such as stripes and grids. To corroborate our conclusion, we validate it using a complexity metric derived from a mathematical model. Bagrov *et al.*[2] proposed a computational method for estimating the structural complexity of natural patterns/images. We compute the multi-scale structural complexity for these four image categories, and the results corroborate our observed ordering, as depicted in Figure 6 (b). This provides some mathematical evidence to support our claim.

## 3.2 Reconstruction on Background

The aforementioned results indicate that the deraining capability can be enhanced by limiting the complexity of the background images used for training. However, utilizing only a restricted set of background images also has drawbacks. While it prevents the network from overfitting rain patterns, it may conversely prompt the network to overfit the limited background images. We also conduct experiments to address this particular concern.

Using the decoupled evaluation metric $E_B$ described in Section 2.2, we are able to assess the reconstruction of the background separately. The results in Figure 7 show that as the number of training images increases, the quality of background reconstruction also improves. Remarkably, training with just 256 background images can already yield a satisfactory background reconstruction performance. Adding more training images beyond this point does not significantly improve performance. These findings are surprising and counter-intuitive. We typically assume that training low-level vision models requires a large number of images. However, our research suggests that training with an excessive number of background images does not necessarily enhance the reconstruction performance but rather exacerbates the model's tendency to overfit rain streaks. Another unexpected result is that a model trained with merely 256 images can already handle most image components. This might imply that the complexity of image components and features for a low-level vision network might not be as high as commonly believed.

**Relationships with large models and large data trends.** We have also noticed recent trends around large models and large data. Many studies point out that larger models (from billions to hundreds of billions of parameters) and larger data can lead to the emergence of capabilities that are previously unimaginable [26]. This seems to contradict the discovery described in this work.

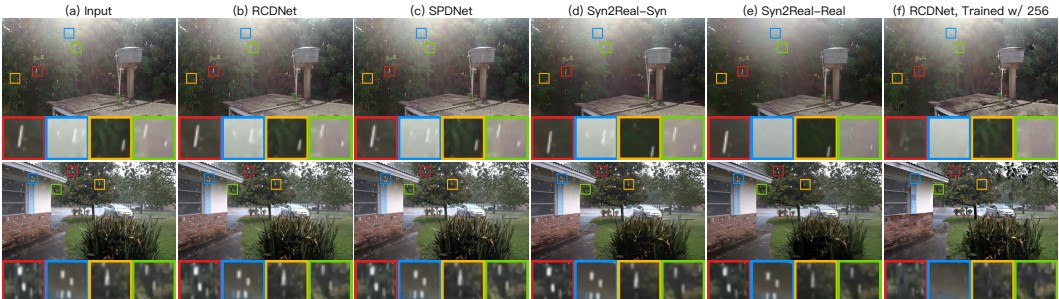

Figure 9: Qualitative results on real-world test images. Zoom in for better comparison.

However, the models that can achieve significant gains from a large amount of data and model parameters are some generative-based models, such as GPT [3] and diffusion models [41]. The large-scale dataset is not the only critical factor to the remarkable ability of these large models. The training method is also very important. Giving a large-scale dataset to a simple task will not make the model generalize to other tasks. In low-level vision, we argue that the task of learning an end-to-end input-to-output mapping via a simple image comparison loss may be relatively easy. In contrast, an important factor for the success of the large language model is its generative pre-training learning method. This learning method enables the model to learn the content rather than the mapping described by the input and output. It is also the core point of this paper that solving the generalization problem requires learning the content of the data, instead of the degradations.

## 4    Implication

This paper does not propose any new algorithms directly, but can provide insights on how to improve the generalization performance of low-level models. Our experiments have yielded three significant practical findings: (1) By limiting the number of background images used in training, the network can focus more on learning the image content instead of overfitting the rain streaks; (2) Enlarging the range of rain streaks in the training set can allow for the use of more background images in training; (3) Surprisingly, a small number of background images can yield satisfactory reconstruction performance. These findings can be directly applied to enhance the generalization capability of existing models with minimal modifications. Our strategy is straightforward: *find a balance between the number of background images and the range of rain streaks in the training set to avoid overfitting.*

Some quantitative results are presented in Table 2. We use three deraining models as baselines (ResNet, SPDNet [68], RCDNet [54]) and demonstrate the power of the proposed strategy. We train our baseline models by using 30,000 background images and medium-range rain. The test set is the R100 dataset [63]. We quantify the deraining and background reconstruction effects according to the decouple evaluation metrics $E_R$ and $E_B$. We also calculate the PSNR performance as a reference. It can be seen that using the existing training methods cannot generalize well to the unseen rain of R100, which is shown by the poor deraining performance in Table 2. However, due to the learning on a large number of images, the reconstruction errors of the baseline models are generally lower. Thus, the PSNR values cannot objectively reflect the effect of rain removal. We reduce the training background images to 64, which is the upper limit of the image number that can make the model generalize under medium-range rain. At this time, the rain removal performance has been greatly improved, but at the cost of background reconstruction performance. By enlarging the rain range and training with more background images, we are able to achieve a trade-off between rain removal performance and background reconstruction.

Figure 8 shows a qualitative comparison of these models under different training objectives. It can be seen that even with the advanced network structure design, the rain removal effects of the baseline models of SPDNet and RCDNet are not satisfactory. Using a larger range of rain can bring limited improvements. In the case of medium-range rain, reducing the background image to 64 significantly improves the rain removal effect and results in unstable image reconstruction. When the rain range is enlarged, and the training background is set to 128 images, the model can show excellent performance in rain removal and background reconstruction. Note that we do not use additional data or improve the network structure throughout the process. We only adjust the training data.

We also present the comparison on real images in Figure 9. In addition, semi-supervised methods [58, 23] have also been used to improve the deraining effect on real images, and we also include the

representative method Syn2Real [66, 67]. Syn2Real-Syn is trained on synthetic data, and Syn2Real-Real is trained on synthetic labelled data and real unlabeled data. Due to the difference in the distribution of rain streaks, the models trained using synthetic data cannot produce satisfactory rain removal effects. When obtaining some real images, Syn2Real-Real can indeed achieve some improvement. However, these improvements are not brought by improving the generalization ability. Because these methods manage to convert "rain outside the training set" to "rain inside the training set". Since data collection is extremely difficult, this method still faces great challenges in practice. Our method improves generalization performance and achieves better results on test images. We also find that artifacts may appear in the background part of the output images. Given that our training utilizes only 256 images without any specific techniques, the outcome, while expected, showcases our prowess in the removal of rain streaks. Indeed, while competing methods might sidestep these artifacts, they fall short in removing rain streaks effectively. We recognize various solutions existing to rectify the artifact issue – from introducing post-processing modules to leveraging this outcome to steer another deraining network. These methods deviate from the interpretability topic of this work, and we hope to discuss technical means to improve generalization performance further in future works.

## 5    Conclusion and Insights

We investigate the generalization problem in deraining networks. While our work focuses on image deraining, our key conclusions can provide insights for the broader field of low-level vision. We argue that the generalization problem in low-level vision cannot be attributed solely to insufficient network capacity. Instead, we discover that existing training strategies do not promote generalization. Networks tend to overfit the degradation patterns in the training set, leading to poor generalization performance for unseen degradation patterns. Moreover, the current need for practical interpretability tools for low-level vision models presents a significant obstacle. These tools are necessary to understand what the low-level model is learning and why it learns this way. This gap between the ideal and the reality compels us to reconsider the importance of interpretability in low-level vision networks. Our work provides a necessary perspective on this issue. These insights also need to be tested and validated in other low-level vision tasks. Our findings highlight the importance of continued research in this area, and we hope that our work will inspire further investigations.

**Acknowledgement**    This work was supported in part by the National Natural Science Foundation of China under Grant (62276251, 62272450), the Joint Lab of CAS–HK, the National Key R&D Program of China (NO. 2022ZD0160100), and in part by the Youth Innovation Promotion Association of Chinese Academy of Sciences (No. 2020356).

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

# Appendix

## A  Other Related Work

### A.1  Image Deraining

Many methods have been proposed to develop state-of-the-art deraining networks. These works include deep networks designs [12, 53], residual networks [13, 32], recurrent networks [40, 64, 62], multi-task [57, 11] and multi-scale designs [25, 14, 65, 69, 59, 52, 70], sparsity-based image modeling [21, 78], low-rank prior [4], model-driven solutions [54, 56], attention mechanism [51, 6, 15], Transformer-based network [8], adversarial learning [29], representation learning [5], semi-supervised [66] and unsupervised learning [9]. Deep learning methods are data-hungry, but collecting rain streaks and background image pairs is challenging. A lot of works have been proposed to synthesize rain streaks with better results. Garg *et al.*[16] first propose a physically-based photo-realistic rendering method for synthesizing rain streaks. Zhang *et al.*[71] and Fu *et al.*[12] use Photoshop software to manually add rain effects to images to build the synthetic paired data. Due to the poor generalization performance of existing methods, models trained on synthetic images were found to be ineffective in real-world scenarios. Some works [63, 72, 55] that have contributed to real-world collected deraining datasets. However, acquiring these datasets is still expensive and cannot solve the problem of poor generalization. There are also works that mention the generalization issue of the deraining models. Xiao *et al.*[60] and Zhou *et al.*[77] attempt to improve the generalization ability of deraining networks by accumulating knowledge from multiple synthetic rain datasets, as most existing methods can only learn the mapping on a single dataset for the deraining task. However, this attempt does not allow the network to generalize beyond the training set.

In addition, semi-supervised methods [58, 23] have also been used to improve the deraining effect on real images, and we also include the representative method Syn2Real [66, 67]. There are some semi-supervised deraining methods [58, 23, 66, 67] are proposed to improve the performance of deraining models in real-world scenarios. When obtaining some real images similar to the test images, these works can achieve some improvement. However, these improvements are not brought about by improving the generalization ability. Their solution is to include real test images in the training set, even if we don't have corresponding clean images. These methods are effective when we can determine the characteristics of the test image. However, more is needed to solve the generalization problem. Because these methods manage to convert "rain outside the training set" to "rain inside the training set". Since data collection is extremely difficult, this method still faces great challenges in practice.

### A.2  Low-Level Vision Interpretability

Next, we provide a detailed review of existing work on low-level visual interpretability. Gu and Dong [19] bring the first interpretability tool for super-resolution networks. Xie *et al.*[61] find the most discriminative filters for each specific degradation in a blind SR network, whose weights, positions, and connections are important for the specific function in blind SR. Magid *et al.*[37] use a texture classifier to assign images with semantic labels in order to identify global and local sources of SR errors. Shi *et al.*[45] show that Transformers can directly utilize multi-frame information from unaligned frames, and alignment methods are sometimes harmful to Transformers in video super-resolution. They use a lot of interpretability analysis methods in their work. The closest work to this paper is the deep degradation representation proposed by Liu *et al.*[33]. They argue that SR networks tend to overfit degradations and show degradation "semantics" inside the network. The presence of these representations often means a decrease in generalization ability. The utilization of this knowledge can guide us to analyze and evaluate the generalization performance of SR methods [34].

## B  Transferability of Limited Training Images

At the end of the main text, we described a method to improve the generalization performance of the deraining network by reducing the number of training background image images. However, this method will overfit the image content when the number of training background images is very small. In the main text, we investigate the risk of reducing the number of training images when testing on the same image category. Recall that with the increase of training images, the reconstruction of the background becomes better. Training with 256 background images can already bring a good background reconstruction effect. Continuing to add training images does not further improve the performance of background reconstruction.

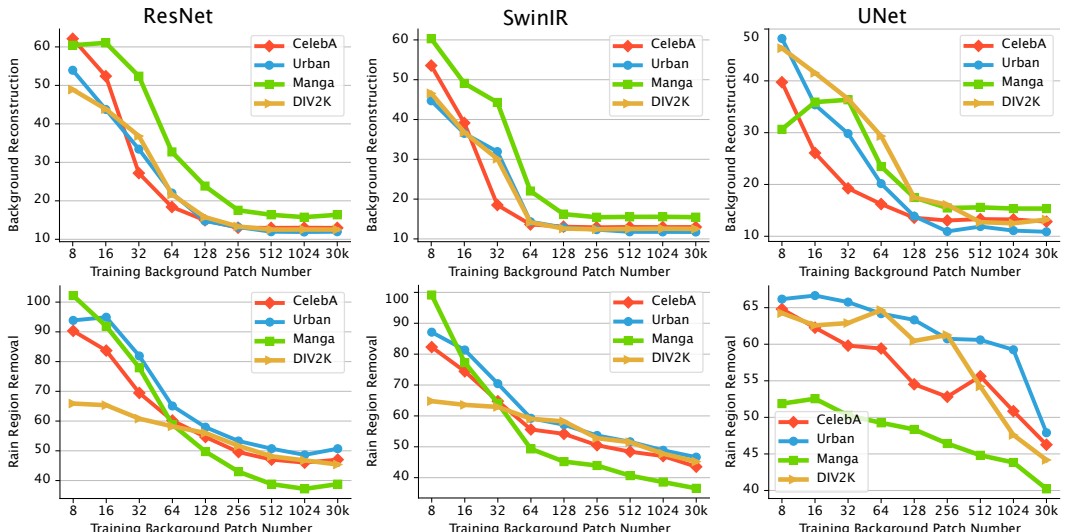

Figure 10: The relationship between the number of training images and their background reconstruction performance (**upper row**) and rain removal performance (**lower row**). The test image set for these six plots is the DIV2K set. We train the model with all four image categories to validate the performance when the image distribution mismatch. For background reconstruction, lower values on the $y$-axis mean better background reconstruction. For rain removal, higher values on the $y$-axis mean better rain removal performance.

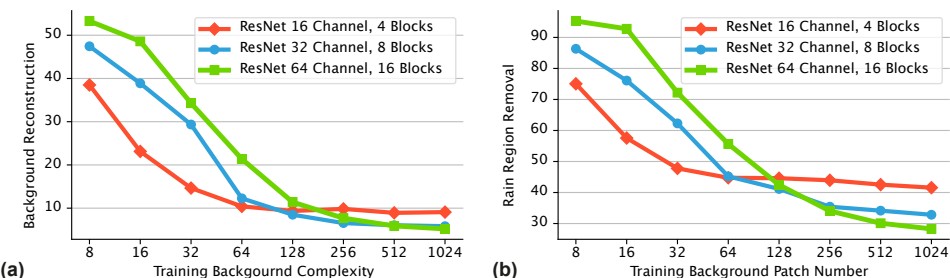

Figure 11: The relationship between the number of training images and their background reconstruction (**a**) and rain removal (**b**) performance. The test image set for these two plots is the DIV2K set. We train the models with different model complexities. For background reconstruction, a lower value on the $y$-axis means better. For the rain removal effect, a higher value on the $y$-axis means better. The test rain patterns are not in the training set. The effect of rain removal at this time reflects the generalization performance.

In this section, we investigate whether the proposed scheme is still robust when the training and testing image distributions are significantly different. We train the models on four image categories and then test them using the DIV2K image category. This simulates the situation when the background image distribution differs from the test set. We observe the behaviour of models trained with different numbers of images. The results are shown in Figure 10. We can draw the following conclusions. First of all, even if the distribution of training background images is different, the models trained using the images of CelebA and Urban categories can still perform similarly to the model trained by DIV2K images. These models can reconstruct background images well when training images reach 256 or more. At this time, the difference in the distribution of these training sets and DIV2K does not bring significant differences. The rain removal effect of these models is also similar. Second, we found that the model trained with the Manga image differed from others. The models trained with the Manga images are generally worse at background reconstruction than other models. Even with larger image numbers, the models trained with the Manga images still cannot achieve similar performance to other models. For rain removal, the model trained with the Manga images also performs the worst. These results are in line with our expectations because the Manga images significantly differ from other images, especially for the low-level image components. Although the other three categories of images

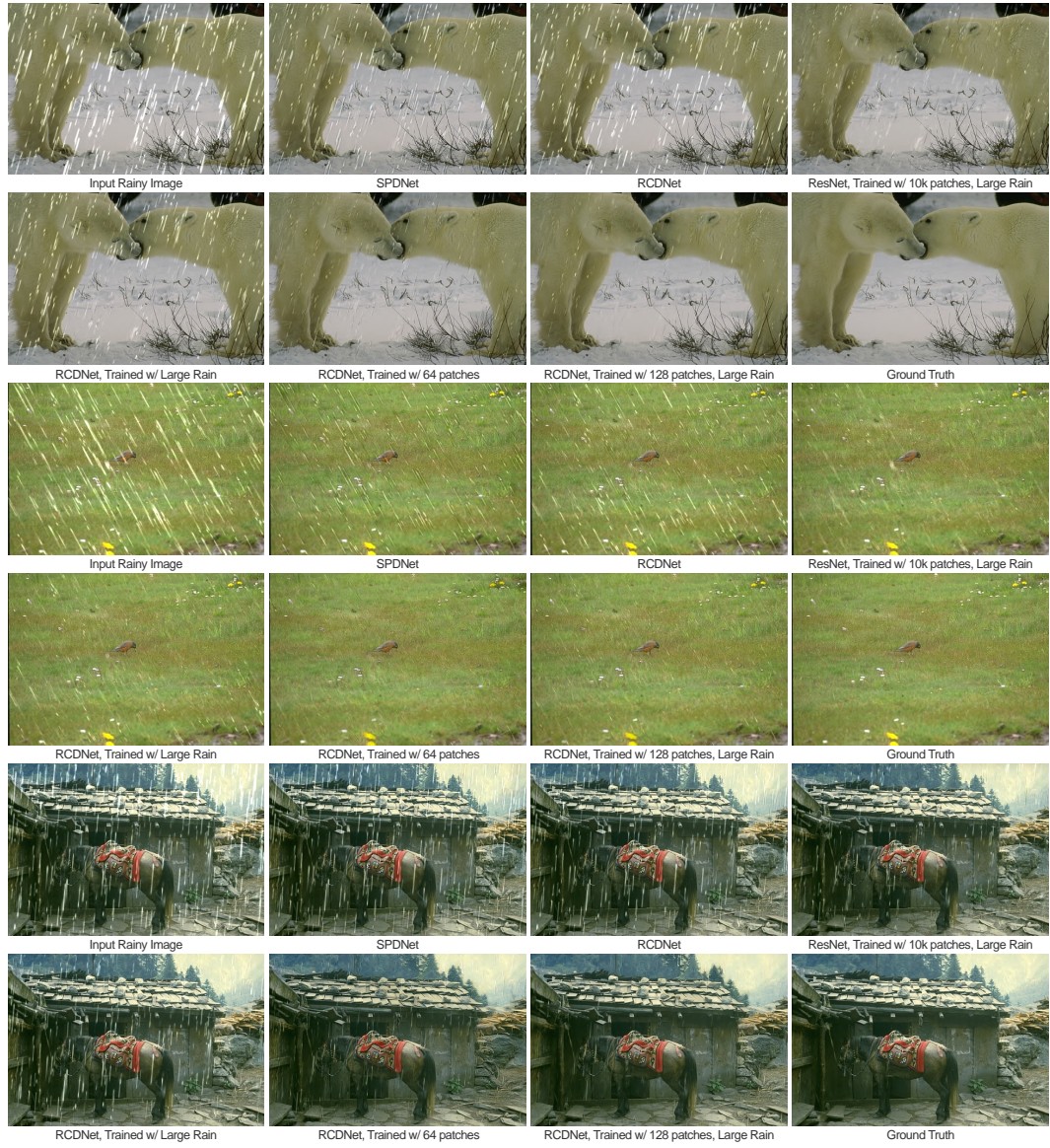

Figure 12: Visualization of the deraining results. Zoom in for better comparison.

are also different from each other in image structure, texture type, and other characteristics, they all belong to the category of natural images. Whereas the Manga images contain artificial textures and edges, which are quite different from other images.

There is a large image reconstruction error when training with images whose distribution is very different from that of the test set images. This is reasonable because, in this case, even using a large number of training images cannot bridge the error caused by the distribution mismatch. We are pleased that the method of training using limited background images is robust to image content to a considerable extent, as long as the training images are natural images. This is consistent with our practice. In practice, we also found that the results are stable as long as more than 256 image images are used for training. There is no significant performance change due to the content of the selected training images.

## C Discussion about Model Complexity

The impact of model complexity on generalization performance cannot be ignored either. In general, models with smaller capacity are less prone to overfitting, although this is in exchange for generally lower performance. We next investigate the validity of our conclusions under models with different complexity. Specifically, we use three different sizes of ResNet architectures and conduct experiments

on the DIV2K dataset, aligning with the experiments presented in Figure 4 and Figure 7. The results are illustrated in Figure 11. One can see from the results that all three models of varying complexities exhibit similar behaviours in rain removal performance. When the training background complexity is on the lower side, the more complex models deliver superior rain removal generalization (as can be seen in the right figure, where higher is better). Conversely, with the increase of the training background complexities, these more complex models encounter larger generalization challenges. More importantly, the inflection point, where this transition occurs, remains consistent irrespective of the model's complexity. This confirms our explanation that within this training framework, the behaviour of the network is primarily determined by the relative complexity between the training background image and the rain degradation. Similar trends are noticeable in the context of background reconstruction. For more complex models, the quality of background reconstruction tends to degrade more when trained with a limited number of images. This can be attributed to the pronounced overfitting tendency of larger models. Interestingly, the rain removal performance is enhanced at this time, indicating the model's focus on image content. As the number of training images increases, the larger models also yield improved background reconstruction, which aligns with our expectations. It is crucial to highlight that the transition in network behaviour remains consistent across models of differing complexities. This reaffirms our conclusion that, within this framework, the network's behaviour is partly steered by the relative complexity interplay between the training background images and the rain degradation.

## D  More Results

We provide more results of different deraining models in Figure 12 and Figure 13. Note that we did not use additional data nor improve the network structure throughout the process. We only adjust the training objective. Although the effect of the output image can be further improved, it shows the practical value and application potential of our conclusions.

## E  Limitation.

Our work mainly takes the deraining task as a breakthrough point, and attempts to make a general summary of the generalization problem in low-level vision. Due to the differences between different low-level tasks, the analysis methods in this paper, especially the fine-grained analysis methods, may not be directly used on some other tasks. However, our work can still bring novel insights to the low-level vision field.

Our work also attempts to improve existing deraining models. But these improvements are based on the simple usage of some key conclusions of our work. Although shown to be effective, we believe that these methods are still far from ideal. We only demonstrate the application potential of the knowledge presented in this work and have no intention of proposing state-of-the-art algorithms or models. Research efforts are still needed to develop more robust deraining algorithms using our conclusions.

## F  Reproducibility Statement

### F.1  Resources

The models used in our work are taken directly from their respective official sources. Our code is built under the BasicSR framework https://github.com/ xinntao/BasicSR for better code organization. The deraining model SPDNet [68] is available at `https://github.com/Joyies/SPDNet`. The deraining model RCDNet [54] is available at `https://github.com/hongwang01/RCDNet`. The training and testing datasets used in our work are all publicly available.

### F.2  Network Training

Due to space constraints, we do not describe our training method in detail in the main text. Here, we describe the training method to reproduce our results. A total of 150 models were involved in our experiments. We used the same training configuration for all models. We use Adam for training. The initial learning rate is $2 \times 10^{-4}$ and $\beta_1 = 0.9$, $\beta_2 = 0.99$. For each network, we fixed the number of training iterations to 250,000. The batch size is 16. Input rainy images are of size 128×128. The cosine annealing learning strategy is applied to adjust the learning rate. The period of cosine is 250,000 iterations. All models are built using the PyTorch framework [39] and trained with NVIDIA A100 GPUs.

### F.3  Availability

All the trained models and code will be publicly available.

# G Ethics Statement

This study does not involve any human subjects, practices to data set releases, potentially harmful insights, methodologies and applications, potential conflicts of interest and sponsorship, discrimination/bias/fairness concerns, privacy and security issues, legal compliance, and research integrity issues. We do not anticipate any direct misuse of our contribution due to its theoretical nature.

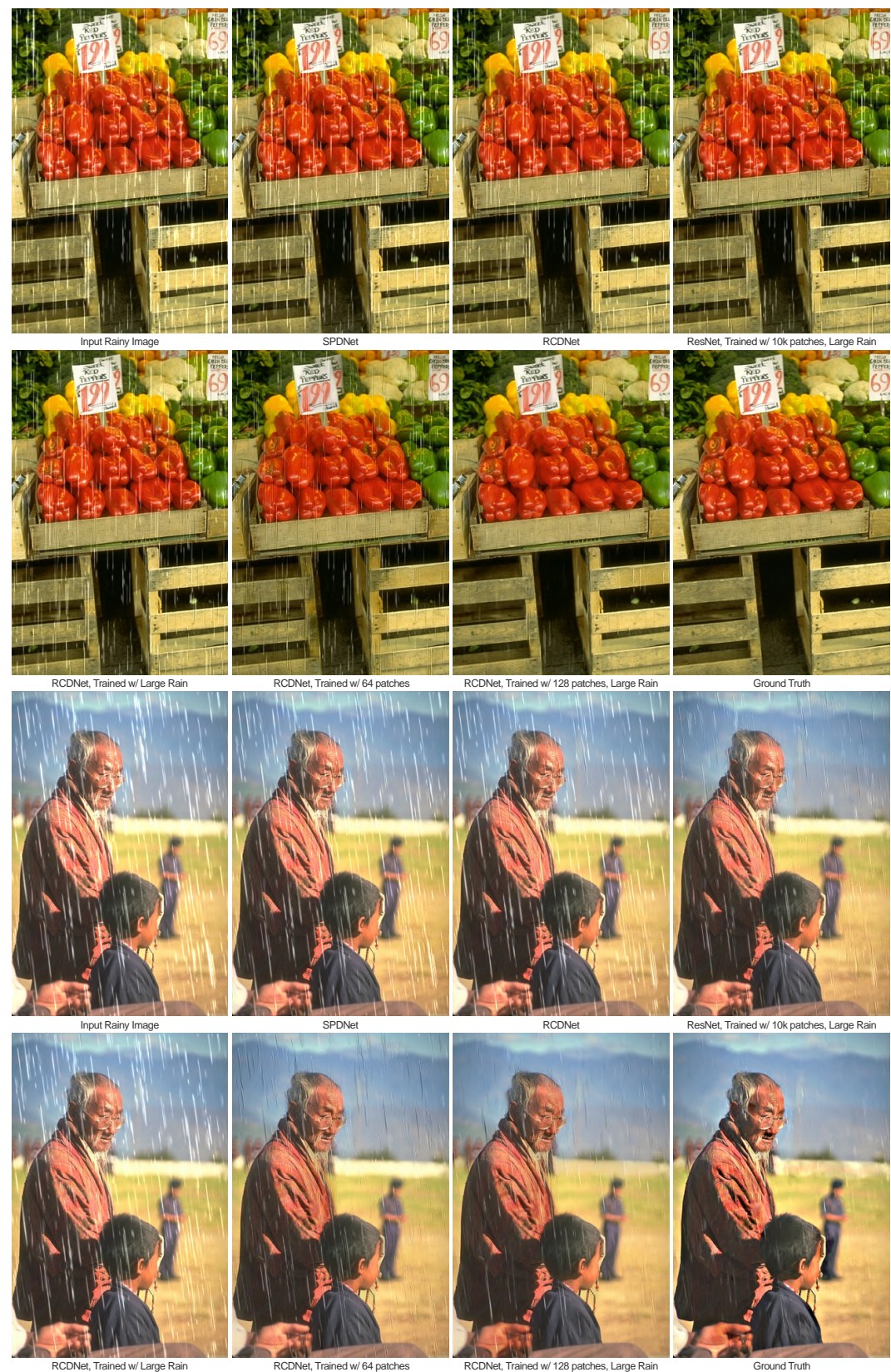

Figure 13: Visualization of the deraining results. Zoom in for better comparison.

