# OpenReview forum: "Networks are Slacking Off: Understanding Generalization Problem in Image Deraining"
_NeurIPS.cc/2023/Conference — NeurIPS 2023 poster_

### Official Review · Reviewer_FkGF · 2023-06-28

**Soundness:** 3 good
**Presentation:** 3 good
**Contribution:** 3 good
**Rating:** 8
**Confidence:** 5

**Summary:**

This paper explores the generalization problem of image deraining. The authors claim that deep networks display a tendency to learn the less complex element in the task of separating image content from additive degradation. Overall, this paper contributes some new ideas and new insight to the community, but there are still some weaknesses that can be further improved.

**Strengths:**

S1. The paper is well-organized and clearly written.

S2. The authors point out the the generalization problem in the context of deraining tasks, which contributes some important ideas for the community. The key findings of this paper left a deep impression on me.




**Weaknesses:**

W1. The author chose two CNN-based methods, RCDNet and SPDNet, to validate the main finding of this paper. I am curious whether the latest Transformer-based image deraining method still has the same conclusion. For example, Chen, et al. "Learning A Sparse Transformer Network for Effective Image Deraining." CVPR, 2023. In fact, the generalization performance of CNN-based and Transformer-based deraining methods is different. I suggest the author could add the experimental analysis of the Transformer-based deraining method.

W2. The authors claim that the relative complexity between the background and rain determines the network behavior. In previous studies, Rain100L and Rain100H are two classic image deraining benchmark datasets that share the same background image. As far as I know, Rain100H (rain is more complex) has better generalization performance than Rain100L. I am curious about how to define the relative complexity between the background and rain. The expression here needs further improvement, which is a bit difficult to understand.

**Questions:**

See the above Weaknesses part.

**Limitations:**

The author has clarified the limitations in the supplementary materials.

---

> ### Author Rebuttal · Authors · 2023-08-09
>
> We thank reviewer FkGF (R#4) for his review. We also thank R#4 for acknowledging our conclusions and contributions. Our response to your concerns is as follows:
>
> `In Weakness 1): "I am curious whether the latest Transformer-based image deraining method still has the same conclusion."`: Thank you for the reviewer's suggestion. In our paper, we primarily utilized SwinIR as a representative of the Transformer networks. Our results indicate that, when trained under the same conditions, SwinIR does not exhibit superior generalization performance compared to CNN-based networks. Given SwinIR's role as a representative method of Transformers and its inherently simple and effective Design design, we infer that Transformers do not inherently offer enhanced generalization capabilities. However, we cannot conclude that no Transformer design can achieve substantial improvements in generalization. There might be newer methods that address the generalization issues of Transformers: reference [3] serves as a case in point. While they also used SwinIR, they introduce the attention and input mask module, significantly enhancing the model's generalization capacity. Based on our tests, the model proposed in [3] is also effective in improving the generalization capability for the deraining task. The paper mentioned by the reviewer has been accepted for CVPR2023. Since it was not publicly published at the time of our submission, we did not discuss or cite it in our initial version. We have added a discussion about this paper in our revised version. We appreciate the reviewer bringing this to our attention.
>
> `In Weakness 2): "As far as I know, Rain100H (rain is more complex) has better generalization performance than Rain100L."`:
> Thank you for the question. Compared to Rain100L, Rain100H features denser rain, larger raindrops, and more obscured background content. The enhanced generalization ability of models trained on the Rain100H dataset might stem from the broader extent of occlusion. This intriguingly aligns with the approach utilized in reference [3]. In that work, they discovered that when training denoising networks, adding a random mask layer to the input image largely transforms the denoising task into an inpainting task. Consequently, in an effort to minimize the loss, the network is compelled to learn background reconstruction. This, in turn, results in improved generalization performance, which is consistent with our paper's findings. Returning to the reviewer's question, due to the more extensive background occlusion by rain in Rain100H, the network focuses more on learning image content rather than raindrop distribution. This might be the reason for its superior generalization capability.
>
> `In Weakness 2): "I am curious about how to define the relative complexity between the background and rain."`:
> Thank you for the question. Admittedly, this is a challenging question to address. Defining a quantitative measure for such complexity is daunting. Complexity isn't solely about the data; the design of the model itself also plays a significant role. Consider a network specifically crafted for learning raindrops or human faces; for such architectures, the "complexity" of learning raindrops or faces might be relatively lower. At this juncture, we cannot provide a mathematically sound measure that quantifies all these elements. However, our approach manifests this relativity in another manner. As the number of images increases, their learning complexity monotonically rises, while the complexity of the rain remains constant. We observed a shift in network behavior throughout this rising process, affirming the existence of relative complexity.
>
> From another perspective, in real-world applications, the complexity of background images always greatly surpasses that of degradation. This accounts for the prevalent generalization problems in low-level vision tasks. Overcoming generalization issues by merely reducing the complexity of background images isn't appropriate. This realization inspires us to develop more sophisticated training strategies and models.
>
> Reviewer can also encouraged to check the new experiment provided in the rebuttal pdf for more information.

---

> ### Author Response · Authors · 2023-08-13
> **Call for Further Discussion**
>
> Dear reviewer FkGF:
>
> We thank you for the precious review time and valuable comments. And Thank you for recognizing our work
>
> We have provided corresponding responses and results, which we believe have covered your concerns. We hope to further discuss with you whether or not your concerns have been addressed. Please let us know if you still have any unclear parts of our work.
>
> Best,
>
> Paper 798 Authors.

---

> > ### Comment · Reviewer_FkGF · 2023-08-15
> > **Response to Authors**
> >
> > Thanks for the author's detailed responses. This paper contributes some new insight to low-level vision interpretability research, and the rebuttal also answered my concern clearly. Besides, it is better to add these analyses in the rebuttal to the released version.

---

### Official Review · Reviewer_w1Er · 2023-06-30

**Soundness:** 3 good
**Presentation:** 3 good
**Contribution:** 3 good
**Rating:** 8
**Confidence:** 4

**Summary:**

This paper proposed to explore the problem of generalization problem in image deraining. The authors find that deep networks tend to overfit the degradation patterns present in the training set, which leads to poor generalization performance for unseen degradation patterns.

**Strengths:**

1. Writing fairly well.

2. The problem is interesting. It is very interesting to explore such findings that better generalization in a deraining network can be achieved by simplifying the complexity of the training data. None of previous works have discussed these issues.

**Weaknesses:**

1. Why did the author create a new dataset without conducting experiments on previous existing benchmarks. The author should explain this issue.

2. Is there any connection between the generalization problem proposed by the author and the interpretability of the low-level vision networks? I am confused about the conclusion of this paper.

**Questions:**

Is the main findings of this paper applicable to other low-level vision tasks? Has the author conducted any extension experiments for other tasks?

**Limitations:**

The authors adequately addressed the limitations.

---

> ### Author Rebuttal · Authors · 2023-08-09
>
> We thank Reviewer w1Er (R#3) for his\her efforts. We also thank R#3 for finding our conclusions interesting and novel. Our response to your concerns is as follows:
>
> `In Weakness 1): "Why did the author create a new dataset without conducting experiments on previous existing benchmarks."`:
> We appreciate the reviewer's question. In this paper, our aim is to showcase specific behaviors exhibited by models when using neural networks to tackle the rain removal problem, achieved through meticulously designed experiments. In previous benchmarks and research settings, all variables were intermingled, making it challenging to isolate elements such as the distribution of rain or the distribution of background images. This complication hindered our past analyzes of model performance. Therefore, in this study, we manually curated our training data and training paradigm, striving to se partly discuss and analyze each variable. We do not intend to create a new benchmark, we just study the behavior of deep low-level vision methods in a purer environment.
>
> `In Weakness 2): "Is there any connection between the generalization problem proposed by the author and the interpretability of the low-level vision networks?"`:
> The issue of interpretability in low-level vision encompasses various aspects. Gu and Dong [11] delved into how deep models in low-level vision utilize input information. Xie et al. [33] investigated from the perspective of specific network filters. Magid and colleagues [19] analyzed low-level vision models based on the errors in their results. Shi et al. [22] studied the alignment modules of video super-resolution networks. Meanwhile, Liu and his team [17] analyzed the semantic degradation in low-level vision.
>
> Analyzing the generalization issues in low-level vision and pinpointing the inherent causes of such issues is one of the important topics in low-level vision interpretability research. However, this question has not yet been explored in existing literature. As the first paper dedicated to explaining generalization problems in low-level vision, this work holds significant value.
>
> `In Question: "Is the main findings of this paper applicable to other low-level vision tasks? Has the author conducted any extension experiments for other tasks?"`:
> Thank you for the question. Indeed, our findings can be applied to many other low-level vision tasks, such as real-world image restoration, which closely align with the RealESRGAN setting [R1], and image denoising. We have conducted very similar experiments, and the conclusions align. A recent paper proves our point [3]. Their experiments were conducted on the denoising task. They found that the existing denoising model does not learn the content of the image (this is consistent with our point of view), causing the generalization performance of the denoising model to fail. Through deliberate masked training, the network is forced to learn the reconstruction of image content, This improves the generalization performance.
>
> The reasons for focusing on deraining as the subject of our study are described in lines L34 to L43 of the paper. To summarize, the primary reason is that the degradation in deraining is relatively simple, and our proposed method allows for a clear separation between deraining performance and background reconstruction performance. In contrast, other low-level vision tasks are challenging to study through quantitative means for controlled variables; conclusions can only be drawn qualitatively based on the outcomes.
>
> [R1] Wang, Xintao, et al. "Real-esrgan: Training real-world blind super-resolution with pure synthetic data." Proceedings of the IEEE/CVF international conference on computer vision. 2021.

---

> ### Author Response · Authors · 2023-08-13
> **Call for Further Discussion**
>
> Dear reviewer w1Er:
>
> We thank you for the precious review time and valuable comments. And Thank you for recognizing our work
>
> We have provided corresponding responses and results, which we believe have covered your concerns. We hope to further discuss with you whether or not your concerns have been addressed. Please let us know if you still have any unclear parts of our work.
>
> Best,
>
> Paper 798 Authors.

---

> > ### Comment · Reviewer_w1Er · 2023-08-18
> > **Response to Authors**
> >
> > Thanks for the author's thoughtful reply. The rebuttal addressed my concerns well. To my knowledge, there is a lack of interpretable research in the field of image rain removal, and this paper opens up a new perspective for the community. I believe that improving the interpretability of the solving process will be beneficial to improve the deraining performance and applications of image deraining methods, which will be important directions for future research.
> >
> > I was originally positive at the paper. When I checked other reviews and the rebuttal, I decided to raise my rating. I believe that the significance, key findings, experimental analysis, and discussion of this work meet the acceptance standards of NeurIPS, and I am inclined to accept this work. This will be an  important pioneering work in the field of interpretable-based image deraining.
> >
> > Best,
> > Reviewer w1Er

---

### Official Review · Reviewer_e5xd · 2023-07-06

**Soundness:** 2 fair
**Presentation:** 3 good
**Contribution:** 2 fair
**Rating:** 3
**Confidence:** 5

**Summary:**

This paper is working on understanding generalization problem in image deraining. Several interesting findings are: 1) training with fewer background images leads to better deraining effect; 2) the relative complexity between the background and rain determines the network behavior; 3) a more complex background set makes it harder for the network to learn.

**Strengths:**

there are some interesting findings in this paper for image deraining for example
1) training with fewer background images leads to better deraining effect.
2) the tradeoff between deraining and background reconstruction.

**Weaknesses:**

1) although there are interesting finding, the overall contribution is limited, i.e., authors didn't provide a good solution to solve the problem. for example for figure 9, although results from author removed rain, but it also generated much more artifacts in the background which probably not good for image deraining task. I would expect more contribution from authors to make this paper accepted by NeurIPS.

2) My feeling is that there is fundamental limit in the current training for image derain, the simulated rain streaks is not real. I wondered wether the current findings from author because of this fundamental limit.

3) some related papers like [1] is not cited or compared.

[1] Jie Xiao, Man Zhou, Xueyang Fu, Aiping Liu, Zheng-Jun Zha, Improving De-raining Generalization via Neural Reorganization, ICCV 2021.

I have carefully read authors rebuttal and still thought the quality of this paper could be improved. So will keep my rating.

**Questions:**

see the weakness part.

**Limitations:**

yes, authors provided some limit of this paper, for example, this paper doesn't propose any new algorithms directly.

---

> ### Author Rebuttal · Authors · 2023-08-08
>
> We thank Reviewer e5xd (R#2) for his\her efforts. Thank you for finding our conclusions interesting. Our response to your concerns is as follows:
>
> `In weakness 1) "the overall contribution is limited, i.e., authors didn't provide a good solution to solve the problem." and "I would expect more contribution from authors to make this paper accepted by NeurIPS."`:
> We respectfully disagree with the R#2's comment. It has been a long time that low-level vision research has predominantly focused on results. Many researchers overlook the fundamental causes of problems, espacially since deep leanring is applied in low-level vision. It seems that any research that doesn't directly yield positive results, cannot be directly applied to algorithm development, or doesn't introduce a new method should neither exist nor be reported or discussed. We are disappointed and frustrated by this prevalent understanding within the field. By this rationale, many seminal papers should never have been accepted. We do not believe that introducing a novel algorithm is a prerequisite for a paper to be accepted by NeurIPS. Our work is devoted to elucidating the underlying causes of the generalization issue in low-level vision, complemented by explorative experiments. The conclusions drawn in our paper can inspire a lot of low-level vision models that are generalizable in real-world applications.
>
> Furthermore, our research has demonstrated, through experimentation, that effects unattainable in current research paradigms can be achieved with just a straightforward approach. Such insights can directly inspire numerous studies. For instance, a recent work [3] which focused on denoising tasks. They found that existing denoising models don't learn image content (which aligns with our views), leading to failures in their generalization capabilities. Through deliberate masked training, they forced networks to learn content reconstruction, thereby enhancing generalization performance. Although the specific methods differ, the underlying approach is to shift training objectives towards understanding image content, rather than overfitting to degradation. This underscores the value of our work. Therefore, we strongly contest the reviewer's critique of our contributions.
>
> `In weakness 1) "for figure 9, although results from author removed rain, but it also generated much more artifacts"`:
> We are confused by the R#2's comment. Figure 9 presents a comparison of our method with existing techniques on real images. In fact, our approach successfully removed the white raindrops while preserving the texture in the background. In contrast, most methods we compared against couldn't effectively eliminate the raindrops. This alone validates the efficacy of our method. We did not observe the artifacts mentioned by the reviewer in the images. We surmise that the reviewer might have mistaken the residual background textures for artifacts. These textures are intrinsic to the background and are not the result of any shortcomings in our outcomes.
>
> `In weakness 2) "My feeling is that there is fundamental limit in the current training for image derain, the simulated rain streaks is not real."`:
> This statement of the R#2 is correct. The generalization problem in low-level vision refers to the performance drop or model failure that occurs when the test degradation is inconsistent with the training degradation. As we explain in the text L316 to L325. There are two main ideas to solve the generalization problem so that the model can be applied in real scenarios. One is to make the training degradation as close as possible to the testing degradation. Existing attempts to propose image synthesis methods, or through real-world data pairs, or unpaired training methods using real data are essentially this idea. Another idea is to improve the generalization performance of the model. Allows a model trained on one type of data to work on a different test data. This is the focus of this paper. In order to study the generalization problem, it is very important to construct the scene of the generalization problem. The rain simulation method we use provides a suitable research scenario for the study of generalization problems. We can use more accurate raindrop simulation methods to improve its effect in real scenes, but this has little to do with the exploration of generalization in this paper.
>
> `In weakness 3) "some related papers like [1] is not cited or compared"`:
> Thanks to R#2 for the question. However, we have discussed the literature mentioned by the reviewers in the supplementary material. In the supplementary material we provide a more detailed literature review. The [32] citation is the paper mentioned by the R#2.

---

> > ### Comment · Reviewer_e5xd · 2023-08-18
> >
> > for figure 9, for the top image when check the wood board we could clear see it quite different from grouthtruth than other methods.
> > for bottom images, when you check the top part of red bricks, middle of blue window, the eaves, it is also clearly worse than other methods.
> >
> > any reason why not compare with [32] and put a highly relevant paper to supplementary materials?

---

> > > ### Author Response · Authors · 2023-08-18
> > > **Response to Reviewer e5xd's Comment**
> > >
> > > Thanks for the reviewer's response.
> > >
> > > ```Figure 9```:
> > >
> > > We appreciate the astute observations by the reviewers about Figure 9. Upon revisiting, we do acknowledge the highlighted shortcomings. Given that our training utilized only 256 images without any specific techniques, the outcome, while expected, showcases our prowess in the removal of rain streaks. Indeed, while competing methodologies might sidestep these artifacts, they fall short in eliminating rain streaks effectively.
> > >
> > > However, we wish to emphasize that these results do not diminish the core contribution of our study. Our paper mainly focuses a counter-intuitive insight, shedding light on it from the lens of low-level vision interpretability. Such an understanding stands poised to reshape the community's grasp on the generalization challenge inherent to low-level vision.
> > >
> > >
> > > Our "implication" experiments provide a new possible way for real-world issues, revealing previously unknown results and potentials. We consciously chose not to present our method as a "novel method" but rather a deep dive into an intriguing phenomenon. We recognize various solutions exist to rectify the artifact issue -- from introducing post-processing modules to leveraging this outcome to steer another deraining network. However, introducing such augmentations might divert from our paper's central theme: a significant revelation about interpretability that holds paramount importance. We think whether the new model will perform better on the benchmark, and how much better it can be, is far less than the significance of our core interpretability findings contribution to the community.
> > >
> > > We still thank the reviewers for pointing out issues with our results. We will be honest about this flaw and highlight the issue of our results in this artifact in the section describing our results in the paper. Yet, we believe labeling our work as having "limited novelty" or worthy of "rejection" based on this alone might be an oversight.
> > >
> > >
> > > ```Compare with [32]```:
> > >
> > > We can, of course, provide comparison results with [32] if the reviewer insist. But we reiterate that our work's essence doesn't lie in benchmarking against state-of-the-art outcomes. As we have repeatedly emphasized, the main contribution of this paper is to demonstrate a counter-intuitive possibility and quantitatively analyze this counter-intuitive result from the perspective of interpretability. Our analysis will largely change the community's understanding of the generalization problem in low-level vision.
> > >
> > > We understand and respect the rain-removal community's pursuit of superior benchmark numbers and the significance of proposing novel and effective methods. But ignoring experiments that don't directly yield positive results or immediately inform algorithm development is concerning and worrisome. Although innovative works may face criticism from many sides, they are not inferior to methods-focused research. Our results show some unexpected findings. It wouldn't be good for our community to overlook them just because they're different from previous rain-removal studies that had top results.
> > >
> > > In response to the reviewer's specific request, current NIPS regulations prevent us from submitting a revised version with a comparison to [32]. But we'd love to include a direct comparison to [32] in the next version. We are confident in the enduring value of our work, no matter how many experiments are added.
> > >
> > >
> > > ```Thanks```:
> > >
> > > Once again, we're grateful for the feedback. Should there be further concerns needing clarification, please let us know. We're committed to ensuring our work's transparency and robustness.

---

> > > ### Author Response · Authors · 2023-08-21
> > > **Further Response to Reviewer e5xd's Comment**
> > >
> > > Dear Reviewer e5xd,
> > >
> > > We have responded to your question. We hope our latest response addresses your concerns.
> > >
> > > We realize that your main concern is paper contribution. But the other reviewers all acknowledged our contributions. Reviewer w1Er commented "**this paper opens up a new perspective for the community**" and "**this will be an important pioneering work in the field of interpretable-based image deraining**". Reviewer FkGF also commented "**This paper contributes some new insight to low-level vision interpretability research**". Even Reviewer UxXt acknowledges our contribution that "**It is interesting and valuable to investigate the reason**" and "**It is interesting to see such study on low level vision**".
> > >
> > > Hope you can consider our reply. Thanks for your efforts.
> > >
> > > Best,
> > >
> > > Paper 798 Authors.

---

> ### Author Response · Authors · 2023-08-13
> **Call for Further Discussion**
>
> Dear reviewer e5xd:
>
> We thank you for the precious review time and valuable comments. We have provided corresponding responses and results, which we believe have covered your concerns. We hope to further discuss with you whether or not your concerns have been addressed. Please let us know if you still have any unclear parts of our work.
>
> Best,
>
> Paper 798 Authors.

---

### Official Review · Reviewer_UxXt · 2023-07-07

**Soundness:** 3 good
**Presentation:** 3 good
**Contribution:** 2 fair
**Rating:** 2
**Confidence:** 5

**Summary:**

This paper makes observations on the generalization problem of deraining networks by conducting extensive experiments, on the perspective of training data. This paper claims to find out some underexplored insights that less training data with less complex background can lead to better rain removal ability at the cost of the background reconstruction ability and that these conclusions can benefit the future research of deraining community. Two new quantitative metrics are proposed to isolate the evaluation of ability of rain removal and background restoration and experiments are based on them.

**Strengths:**

1. It is interesting and valuable to investigate the reason why existing deraining methods fail to generalize to unseen rain streaks, from the perspective of training data.

2. It is reasonable to separately evaluate the ability of rain removal and background restoration of deraining models, for the two parts both have influence on the final restoration performance.

3. It is good to conduct extensive experiments to validate the assumptions and useful for those new researchers to deraining tasks.

4. The paper is easy to follow.

**Weaknesses:**

Despite the interesting motivation and efforts on studying the generalization problem of deraining methods, there are some basic problems for this paper.

1. It is negative for the paper to hold the opinion that the background restoration quality can be sacrificed to get better rain removal ability. It is opposite to the practice of most image restoration tasks, i.e., keeping the degradation-free areas unchanged. Deraining, which consists of removing the rain streaks AND restoring those lost background details occluded by these rain streaks, is special that most part of the image is actually unaffected given only rain streaks exist and there is no rain accumulation. Thus, the objective of deraining is to remove as many rain streaks as possible when keeping the background unchanged. It agrees with the loss functions used to train models, that keeping the output more similar to the ground truth. The observation found in this paper, is actually not the deraining models overfit to the pattern of rain streaks, rather that models try to keep the background unchanged and do nothing to the image if rain streaks are not recognized. Training with less data indeed makes models to remove more rain, but also makes them to remove background details. That's why we need more powerful models to train on more diverse data to get better performance on real cases.

2. The higher value of proposed metric 'Rain Removal Performance' doesn't mean the better ability of rain removal. As mentioned above, the goal of deraining is to remove the rain streaks and restore the occluded background textures. Thus, simply calculating the difference between between the rain region of rainy and output images, cannot directly tell the ability of removing rain. For example, in the extreme case, the rain removal ability is 'the best' when the rain region is all set to zeros because the difference between the zeros and white rain is maximized, but it is unwanted for deraining. In addition, the difference between the rainy and ground truth in rain regions is different for different backgrounds. Therefore, it cannot directly tell the ability of rain removal by the relative value of the proposed metric.

3. The lower value of proposed metric 'Background reconstrution' doesn't mean the better ability of background reconstrution. Instead, it can only means that the models do less harm to background details when removing rain streaks. To be more specific, the reconstructed background only lies within the rain region rather than the unaffected background areas. Together with the above point, the metric of 'Rain Removal Performance' is a compoistion of the ability of removing rain and reconstructing background. Thus, the proposed two metrics acturally do not isolate the evaluation of removing rain and reconstructing background. Additionally, it can be found that only a certain amount of background images like 256-1k can help models to learn to keep unaffected areas unchanged, which may be good for future data synthsis.

4. The rain region is obtained by thresholding to calculate the new metrics. It is weird because the experiments are conducted on full synthetic data and the rain is manually superimposed on clean images, which means the accurate rain areas are already known in advance. The threshold value can also have negative effects on the correctness of observations.

5. It is infeasible to make more conclusions than that L1 loss for training is not the optimal choice when the whole training is based on L1 loss. Training with L1 loss aims to tell models to produce more similar outputs to ground truth pixel-wisely and evaluating the performance with PSNR and SSIM is persuing better visual quality. While this paper proposes new evaluation metrics and can only prove that L1 loss cannot help models to remove more rain at the cost of removing background details as expected by the paper. It is more important to make real solutions based on the proposed metrics to guide the training of models to get better results. But it is not considered in this paper.

6. It is simple to make seem-to-be-valuable observations via various experiments and such efforts should be respected. However, such conclusions can also be harmful to researchers who are new to this area if no absolute therories supports such insights. Authors also knows that better background preservation can lead to higher PSNR but remove less rain streaks as described in Sec 2.2. The marching direction of the community should be removing more rain streaks and handle more types of rain while keeping the background textures unchanged rather than removing more rain at the cost of background quality. It is recommended that authors also examine the generalization problem from the perspective of model complexity, and you may find that less complex models tends to remove more rain streaks at the cost of removing more backgrounds rather than doing nothing like more complex models.

7. Let's make a step backward, assuming these observations in this paper are insightful, the strategy proposed by authors that trying to find a balance of complexity of background and rain streaks should be applicable to all deraining models, no matter what architectures and complexities they have. However, the quantitative and qualitative results show that RCDNet may be the only benefited model of the proposed strategy. Thus, it is not a general solution that less training data brings better balance and generalization.

8. The selected SPDNet and RCDNet, etc. are actually not the state-of-the-art models in deraining. There are many more powerful deraining-oriented models or general backbone models in both CNN-based and Transformer-based architectures these years. It is more insightful to examine how these new powerful models behave when the training data of background images is decreasing.

9. For the second implication in Sec 4., is there any chance that larger rain range occludes more background areas and models need more data to learn how to keep unaffected regions unchanged and restore those occluded ones? Different perspectives can lead to various conclusions based on these experimental results.

10. As the LLM and SAM models in NLP and CV areas show up, it is more likely that surprisingly large amounts of training data is the key to generalization problem. However, it is hard to get pair-wise ground truth for so many low-level image restoration tasks, thus it is also valuable to study the possibility of training models to be better with proper data. But the insights claimed in this paper seem to be normal observations when conducting experiments. Most importantly, authors do not give any new solutions based on these observations like new loss functions. Actually, it is promising to design new loss functions in the spirit of isolation of removing rain and restoring background.

**Questions:**

See the weaknesses above.

It is interesting to see such study on low level vision which is underexplored, but still, the observations made in this paper have some problems. I disagree that the background quality can be the cost of removing more rain, which is against the purpose of image restoration, because usually a less complex method tends to remove both rain and background due to less capacity.

The isolation of evaluation of ability of removing rain and restoring background also fails. Authors should reconsider their definition of reconstructing background. For deraining, the problematic areas are only rain regions rather than those unaffected background areas. The removal of rain and reconstruction of background take place in the same area while keeping the unaffected background unchanged is also important. Previous models years ago cannot handle them at the same time due to their inferior model capability and that's why recent models doing nothing to rain and keeping background tidy can get higher PSNR.

For a better improvement of this paper, I recommend authors to try newly proposed powerful models and find out what the difference is. The perspective of model complexity is also important when discussing the generalization problem.

**Limitations:**

NA.

---

> ### Author Rebuttal · Authors · 2023-08-08
>
> We thank Reviewer UxXt (R#1) for his\her efforts. We next responded to R#1's comments. To save space, we set some abbreviations: generalization (GNL), background (BG).
>
> R#1's resistance to data reduction likely arises from a misinterpretation of our intentions. Through experiments, we aim to challenge prevailing notions of low-level vision GNL. We maintain that not all future methods necessitate data reduction for superior GNL performance. Moreover, we do not claim there is a trade-off between BG reconstruction and deraining effects. As an interpretability study, we neither propose a universally applicable method nor implies the practical viability of the method used in this paper. Our perspectives are neither "negative" nor "harmful". In contrast, our view is insightful and worth discussing.
>
> Conducting experiments to shed new light on the GNL issue is very valuable for the community. While there is an undeniable need for more robust models and efficient loss functions, our results have show interesting and important phenomena. We argue that exploring the causes of the GNL problem holds more importance than proposing a new method or metric, particularly in an era where studies investigating low-level vision interpretability are limited. We take pride in this contribution.
>
> `Weakness 1`: The R#1's statement is common in the existing literature, and it is also the "unproven acknowledgment" mentioned in our paper. But, how does the model recognize the BG to keep and the rain to remove? The prevailing academic assumption posits that the model automatically opts to learn the BG. This paper's insight is to challenge this belief through experimentation. Maintaining an "unchanged BG" necessitates identifying either rain or BG, which is where the GNL issue originates. The R#1's statement, "do nothing...if not recognized" implies R#1's belief that current models identify rain, and their inability to generalize stems from failed rain recognition, thereby contradicting R#1's own critique of our work.
>
> `Weakness 2 and 3`: These metrics are proposed for separate quantitative studies of rain removal and BG reconstruction, and we do not purport them as universally applicable. (1) For deraining, given rain's usual high brightness, this metric can reflect rain removal efficacy. While susceptible to corner cases, we've provided visual results that corroborate our metrics' effectiveness. The corner case mentioned by R#1 do not undermine this metric. Hypothetically, if a method outputs an entirely black in rain part, it has technically "removed" the rain. (2) Regarding BG reconstruction, including rain-affected areas in computations would allow the deraining effect to influence BG reconstruction evaluation. This violates our original intention of studying rain removal and BG reconstruction separately. The deraining effects would be reflected in whole-image performance indicators, and is not what this paper trying to do.
>
> `Weakness 4`: Understandably, we know the precise location of raindrops, and the non-zero part of the raindrop image should represent the background. However, the raindrop synthesis method results in a broad non-zero area surrounding the raindrops, causing the mask's range to exceed actual raindrop dimensions. Thresholding can mitigate this issue. Given our low threshold, this process doesn't impact our conclusions.
>
> `Weakness 5 and 6`: Please see the rebuttal PDF.
>
> `In weakness 7 "...RCDNet may be the only benefited model..." and in weakness 8 "The selected...are not SOTA"`: We respectfully disagree with your comments. Our results consistently show that both Unet, ResNet and Transformer SwinIR gain from the described training method. Almost all experiments speak to these facts. RCDNet and SPDNet serve as additional models to align with existing deraining research. As a paper centered on elucidating the deraining model's generalization mechanism, surpassing existing state-of-the-art results isn't requisite. Our research encompasses various models from existing literature, including the classic Unet-shaped network, the CNN network, and SwinIR. Thus, our conclusions exhibit considerable widely validity. Although employing newer models or examining all models would be ideal, it is unfeasible for a 9-page paper.
>
> `Weakness 9`: This is a good question. First of all, we are aware that the range of actual occlusion by rain may have a little influence on the experiments. Therefore, in different ranges of rain settings, the number of raindrops is the same to reduce the impact caused by occlusion. The reviewers also mentioned that due to occlusion, the model is forced to learn more complex BG reconstruction. In fact, according to our insight, if the model is forced to learn to reconstruct the BG, this also improves the GNL ability of the model. A recent paper proves our point [3]. Their experiments were conducted on the denoising task. They found that the existing denoising model does not learn the content of the image (this is consistent with our point of view), causing the GNL performance of the denoising model to fail. Through deliberate masked training, the network is forced to learn the reconstruction of image content, This improves the GNL performance.
>
> `Weakness 10`: The large data is not the only key to the amazing ability of large models. The training method is also very important. Giving a large data to a simple task will not make the model generalize to other tasks. In low-level vision, we argue that the task of learning an end-to-end input-to-output mapping via a simple image comparison loss may be too simple. An important factor for the success of the large language model is its generative pre-training learning method. This learning method enables the model to learn the content of the image, rather than the mapping described by the input and output. This is also the core point of this paper, that solving the GNL problem requires learning the content of the image, not degradation.

---

> ### Author Response · Authors · 2023-08-13
> **Call for Further Discussion**
>
> Dear reviewer UxXt:
>
> We thank you for the precious review time and valuable comments. We have provided corresponding responses and results, which we believe have covered your concerns. We hope to further discuss with you whether or not your concerns have been addressed. Please let us know if you still have any unclear parts of our work.
>
> Best,
>
> Paper 798 Authors.

---

### Author Rebuttal · Authors · 2023-08-09

Reply to all reviewers and area chairs for short abstracts


Dear Reviewers and Area Chairs,
We thank all reviewers and regional chairs for their valuable time. We sincerely appreciate all comments. They gave positive, high-quality comments to our paper and provided a lot of constructive feedback.

We are encouraged by:
1. **All reviewers agree that our findings and conclusions are interesting.**
2. Reviewers UxXt, reviewer w1Er, and reviewer FkGF **agree that our work is valuable and contributes to the field.** Reviewer w1Er commented "None of the previous works have addressed these issues"
3. Reviewers UxXt, reviewer w1Er and reviewer FkGF **agreed that our paper is well written and easy to understand.**

Our research aims to uncover the root causes of low-level vision challenges, with a particular focus on the generalization problem that we study experimentally. But for a long time, low-level vision research only emphasized benchmark results. Many researchers are not concerned with the root cause of the problem and thus ignore interpretability research, especially since deep learning is applied to low-level vision. We noticed that some reviewers were confused about whether our paper gave a way to surpass SOTA and the final performance metrics. It seems that any research that does not directly produce positive results, is not directly applicable to algorithm development, or does not introduce new methods should not exist, and should not be reported or discussed.

**We would like to point out that there should not be only one kind of paper appearing in the low-level vision community.** As a low-level vision interpretability research, this paper has a unique structure. The way to contribute to the field is also different from other papers that focus on novel methods. Hope the reviewers will again consider this paper as a contribution to the low-level vision science. We have replied to each reviewer individually to address any comments. We also provide experiments on model complexity.

We look forward to addressing any concerns the reviewers had. I noticed that we can continue discussions on the Openreview system for some time to come. If the reviewers has any questions, we would be happy to continue the discussion.

Once again we thank all reviewers and regional chairs!

Paper 798 author.

---

### Author Response · Authors · 2023-08-16
**A Thanks Letter and Call for Discussion**

Thanks to reviewers FkGF and w1Er for they have responded to our rebuttal. We thank the two reviewers for their recognition of our work. To avoid disturbing emails, we will not reply to your comments individually. Thanks for your time and effort!

Reviewers UxXt and e5xd, For all your concerns, we have provided responses. Are there any deficiencies in our rebuttal? Whether the corresponding responses and results we provide cover your concerns? The discussion period ending date is August 21. Please let us know if you have any unsolved or other concerns.

Once again we thank all reviewers and area chairs!

Paper 798 author.

---

> ### Author Response · Authors · 2023-08-21
> **Last Chance for Discussion and A Summary**
>
> We thank all reviewers and area chairs for their valuable time and comments. The reviewer-author discussion deadline set by NeurIPS is drawing to a close. After discussing with reviewers and providing more clarifications/results/analyses, we would like to give a brief response.
>
> Reviewer FkGF, Reviewer w1Er all hold a positive side for our work. Reviewer w1Er commented "**this paper opens up a new perspective for the community**" and "**this will be an important pioneering work in the field of interpretable-based image deraining.**". Reviewer FkGF also commented "**This paper contributes some new insight to low-level vision interpretability research**". After the rebuttal, they raised the already positive paper score to “Strong Accept”. This speaks to their willingness to see the paper accepted.
>
> Reviewer e5xd holds negative rating at the first time. Reviewer e5xd contributed to the discussion, and we have also provided updated explanations. Hopefully our further discussion will address the reviewer's concerns.
>
> Reviewer UxXt also holds negative rating at the first time. But he\she lowered the rating **silently**, **without stating any reason**, and **without engaging in any discussion**. Not to mention we have sent several calls for discussions. We believe it is professional to participate in the discussion before the ending date and justify the rating.
>
> Once again we thank all reviewers and area chairs!
>
> Paper 798 author.

---

### Decision · Program_Chairs · 2023-09-21

**Decision:**

Accept (poster)

**Comment:**

The reviews for this paper were extremely polarized, even after rebuttal and discussions. The two reviewers strongly in favor argued that the novel insights should outweigh the non stellar empirical results. The other two reviewers argued that good empirical results are necessary (e.g. in background reconstruction). I side with the favorable authors as good insights are valuable to the community, although I acknowledge that the empirical results are also important. I encourage the authors to take feedback from all reviewers and incorporate them into the final version.